# Comparison of Biological Activities and Protective Effects on PAH-Induced Oxidative Damage of Different Coffee Cherry Pulp Extracts

**DOI:** 10.3390/foods12234292

**Published:** 2023-11-28

**Authors:** Weeraya Preedalikit, Chuda Chittasupho, Pimporn Leelapornpisid, Siriporn Potprommanee, Kanokwan Kiattisin

**Affiliations:** 1Doctor of Philosophy Program in Pharmacy, Department of Pharmaceutical Sciences, Faculty of Pharmacy, Chiang Mai University, Chiang Mai 50200, Thailand; weeraya.pr@up.ac.th; 2Department of Cosmetic Sciences, School of Pharmaceutical Sciences, University of Phayao, Phayao 56000, Thailand; 3Department of Pharmaceutical Sciences, Faculty of Pharmacy, Chiang Mai University, Chiang Mai 50200, Thailand; chuda.c@cmu.ac.th; 4Faculty of Pharmacy, Chiang Mai University, Chiang Mai 50200, Thailand; pimporn.lee@cmu.ac.th (P.L.); siriporn_pot@cmu.ac.th (S.P.)

**Keywords:** coffee cherry pulp, Soxhlet extraction, antioxidative compounds, anti-aging, polycyclic aromatic hydrocarbons, air pollution

## Abstract

Polycyclic aromatic hydrocarbons (PAHs) are the main toxic components of ambient air particulate matter (PM), causing oxidative damage to the skin and ultimately resulting in skin aging. This study was conducted to determine the anti-oxidant, anti-aging properties and protective effects of the extracts of coffee cherry pulp (*Coffea arabica* L.), which is a by-product of the coffee industry, against the oxidative damage induced by PAH exposure in human epidermal keratinocytes (HaCaT). Three different techniques were used to extract the coffee cherry pulp: maceration, Soxhlet and ultrasonication to obtain CCM, CCS and CCU extract, respectively, which were then compared to investigate the total phenolic content (TPC) and total flavonoid content (TFC). The chemical compositions were identified and quantified using high-performance liquid chromatography (HPLC). The results demonstrated that Soxhlet could extract the highest content of chlorogenic acid, caffeine and theophylline. CCS showed the significantly highest TPC (324.6 ± 1.2 mg GAE/g extract), TFC (296.8 ± 1.2 mg QE/g extract), anti-radical activity against DPPH free radicals (98.2 ± 0.8 µM Trolox/g extract) and lipid peroxidation inhibition (136.6 ± 6.2 µM Trolox/g extract). CCS also showed the strongest anti-aging effects based on collagenase, elastase, hyaluronidase and tyrosinase inhibitory enzymes. In addition, CCS can protect human keratinocyte cells from PAH toxicity by increasing the cellular anti-oxidant capacity. This study suggests that CCS has the potential to be used as a cosmetic material that helps alleviate skin damage caused by air pollution.

## 1. Introduction

Skin aging is a complicated interaction of biological, physical, and biochemical processes influenced by intrinsic and extrinsic factors, resulting in a progressive loss of structural integrity and physiological function [1]. For decades, extrinsic skin aging was influenced by a range of environmental factors, such as exposure to sunlight [2], tobacco smoking [3] and air pollution, especially polycyclic aromatic hydrocarbons-rich particulate matter that is capable of activating aryl hydrocarbon receptor (AhR) signaling in the skin [4,5]. However, growing evidence suggests that air pollution plays a substantial role in skin aging [6,7,8]. 

Particulate matter (PM) is a typical indirect indicator of air pollution and one of the most serious issues affecting air quality and climate change. These particles with an aerodynamic diameter of less than 10 microns (PM_10_) or 2.5 microns (PM_2_._5_) have become a strong hazardous pollutant [9]. The physical and chemical features of PM vary depending on their location. However, common chemical elements found in PM consist of polycyclic aromatic hydrocarbons (PAHs) [10]. PAHs could function as a distinct component of PM that strongly activates AhR expressed in keratinocytes and melanocytes to promote ROS [11,12,13]. Several studies have shown that ROS-producing PAH can cause protein denaturation, lipid peroxidation and DNA mutation. Consequently, this phenomenon leads to oxidative damage and the functional impairment of skin cells, which is positively correlated with skin aging processes characterized by hyperpigmentation, deep skin wrinkles and skin laxity [14,15,16,17].

Prolonged skin exposure to the pollutant leads to increased ROS production, activating dermal enzymes such as collagenase, elastase and hyaluronidase [18,19,20]. The primary function of these enzymes is to regulate the degradation and breakdown of collagen, elastin and hyaluronic acid, which are the most abundant extracellular matrix (ECM) components in the dermis. Collagen and elastin help to retain skin elasticity and flexibility, contributing to youthful and healthy aesthetic attributes [21,22]. Hyaluronic acid, another abundant component of ECM, is involved in hydration, helping to retain moisture and influence the skin’s moisture ability [23]. Apart from that, tyrosinase is the rate-limiting enzyme in melanin synthesis, and its overproduction leads to melanin accumulation and numerous skin disorders, such as hyperpigmentation, freckles and age spots [24]. Therefore, inhibiting degradative enzymes including collagenase, elastase, hyaluronidase and tyrosinase is necessary to protect skin from aging and hyperpigmentation [25].

Recently, a growing trend has been observed toward using active ingredients sourced from natural plant extracts, demonstrating negligible adverse effects. Arabica coffee (*Coffea arabica* L.), mostly cultivated in the northern highland regions of Thailand, has become one of the most well-known commercial coffee species [26]. Coffee cherry pulp is the main by-product of primary coffee production, which is known as the wet processing method. This process involves the removal of the mucilaginous layer, or pulp, from the seed [27]. The coffee cherry pulps contain an abundance of phytochemical constituents, particularly polyphenols and alkaloids, which are likely responsible for their potential health benefits as an anti-oxidant [28,29]. The Arabica coffee cherry pulp is abundant in caffeine and polyphenols with chlorogenic acid being the primary phenolic compound [30]. Other compounds have also been identified although in minor amounts, namely, trigonelline, theobromine, theophylline, protocatechuic acid, gallic acid, quercetin, (+)-catechin, (-)-epicatechin, procyanidin and rutin [31,32,33]. Chlorogenic acid has a high bio-availability among humans and displays a variety of significant biological functions including anti-oxidant and anti-inflammatory, antimutagenic, anticarcinogenic and DNA damage protective effects which involve scavenging ROS [34,35]. Caffeine possesses anti-oxidant activity subsequently protecting individuals from health problems connected to oxidative stress. Varma et al. [36] have suggested that caffeine has the ability to prevent oxidative stress, consequently exhibiting significant efficacy in inhibiting the galactose formation of cataracts. Vignoli et al. [37] reported that the anti-oxidant activity of caffeine is dose dependent as higher levels of caffeine correspond to increased anti-oxidant activity.

Several research studies have reported that coffee cherry pulp has potent capabilities resulting from different extraction methods. The hot water extraction of coffee cherry pulp, rich in chlorogenic acid and caffeine, showed strong total phenolic content and anti-oxidant activity [38]. As an ultrasonic-assisted extraction method, vacuum drying at 110 °C removed moisture from wet coffee pulp and was employed to obtain coffee cherry pulp methanolic extract with high amounts of chlorogenic and caffeine that exhibited strong anti-oxidant capacity measured by DPPH and ABTS assays [39]. The coffee cherry pulp in fractionated ethyl acetate solvent demonstrated potential anti-oxidant, anti-tyrosinase, and anti-aging properties by slightly suppressing matrix metallopeptidase-2 (MMP-2) and showing the potential for preventing hyaluronic acid degradation [40]. Furthermore, much research has reported that aqueous or ethanolic coffee pulp extracts containing chlorogenic acid and caffeine as the primary compounds indicated a high total phenolic content and potent anti-oxidant activity [41,42,43]. Nevertheless, many bio-active compounds such as chlorogenic acid are unstable and susceptible to decomposition, especially when exposed to high temperatures or light [44]. Researchers have also explored the potential application of novel extraction techniques such as microwave-assisted extraction, ultrasonic-assisted extraction, pressure-assisted liquid extraction and supercritical CO_2_ extraction compared with conventional solvent extraction methods such as Soxhlet and maceration because of their various advantages [45]. These constitute alternative extraction methods employing environmentally friendly solvents such as ethanol and water, resulting in an extract that is abundant in bio-active compounds [46]. 

Many studies examined the impact of different extraction methods and solvents on the yield of bioactive compounds and anti-oxidant activities in coffee cherry pulp. However, there is a lack of research on simple extraction methods in relation to the biological properties and protective effects of coffee cherry extracts against PAH cytotoxicity by inhibiting ROS production in keratinocytes for potential use in cosmeceutical applications. Therefore, this study aimed to determine the potential of three extraction methods for obtaining cosmeceutical compounds from coffee cherry pulp. The ethanolic coffee cherry pulp extracts from different extraction methods, including maceration, Soxhlet, and ultrasonic-assisted extraction, were compared in terms of their biological properties such as anti-oxidant, anti-aging and antihyperpigmentation activities as well as protective effects against oxidative damage caused by the PAH in HaCaT cells regarding cosmeceutical application.

## 2. Materials and Methods

### 2.1. Reagents

Standard reference material PAH (CRM47930, QTM PAH-Mix, 2000 μg/mL) was purchased from Sigma-Aldrich (St. Louis, MO, USA). DPPH was purchased from Fluka (Buchs, Switzerland). Folin–Ciocalteu reagent and 2,2’-azobis-(2-amidinopropane dihydrochloride; AAPH) were purchased from Merck (Darmstadt, Germany). The 6-hydroxy-2,5,7,8-tetramethyl chroman-2-carboxylic acid (Trolox), chlorogenic acid, caffeine, theophylline, kojic acid, epigallocatechin-3-gallate (EGCG), linoleic acid, bovine serum albumin (BSA), 3-(4,5-dimethylthiazolyl-2)-2,5-diphenyl tetrazolium bromide (MTT), and 2,7-dichlorodihydrofluorescein diacetate (H_2_DCFDA) were purchased from Sigma-Aldrich (St. Louis, MO, USA). Dulbecco’s Modified Eagle Medium (DMEM), fetal bovine serum (FBS) and penicillin–streptomycin were obtained from Gibco Life Technology (Thermo Fisher Scientific, Waltham, MA, USA). Ethanol and dimethyl sulfoxide (DMSO) were purchased from Labscan Asia Co., Ltd. (Bangkok, Thailand). For enzyme and substrate substances, *Clostridium histolyticum* collagenase (EC.3.4.23.3), porcine pancreatic elastase (PE–E.C.3.4.21.36), bovine testis hyaluronidase (E.C.3.2.1.3.5), synthetic peptide of N-[3-(2-furyl) acryloyl]-Leu-Gly-Pro–Ala (FALGPA), N-succinyl-Ala–Ala–Ala–p-nitroanilide (AAAPVN), hyaluronic acid, and tyrosinase from mushroom were purchased from Sigma-Aldrich (St. Louis, MO, USA).

### 2.2. Plant Preparation and Extraction

Arabica coffee cherries were harvested in December from Doi Chang in Chiang Rai Province, Thailand. To prepare coffee cherry pulp, the coffee fruits were washed and the coffee beans were removed from the pulp using a machine separator. Then, the coffee cherry pulps were dried in a hot air oven (Universal oven Memmert UN 55, Schwabach, Germany) at 50 ± 2 °C to reduce the moisture content until reaching a constant weight and then pulverized into powder using a blender (600 W, Viva Collection Blender Phillip, Bangkok, Thailand). The 95% ethanol was used as a solvent extraction with a solid-to-liquid ratio of 1:5 g/mL. These parameters were designed to be constant across all extraction operations to provide uniform results and coherence for comparing three different extraction methods as follows: maceration was performed three times by adding 50 mL of 95% ethanol to 10 g of plant material in a flask. The soaking material was left for 24 h at room temperature [41]. Soxhlet extraction was performed according to the method of Gligor et al. [47] with a few modifications. Altogether, 10 g of plant material was weighed to 50 mL of 95% ethanol in a Soxhlet extractor thimble and placed in the extraction apparatus. The heating plate temperature was adjusted to 210 °C to reach the boiling point of ethanol (75 to 80 °C) for 60 min. Ultrasonic-assisted extraction was performed according to the method of Gligor et al. [47] with a few modifications. This method was performed using an ultrasonic bath (Elmasonic S 100 H, Singen, Germany), operating at a frequency of 37 kHz (220 to 240 V, 550 W) at 50 ± 5.0 °C for 60 min. To maintain the temperature, monitoring temperature every 15 min was conducted during the whole extraction process. After each extraction procedure was completed, insoluble residue was removed by the filtration through the filter paper (Whatman No.1, Kent, UK). The filtrate was concentrated using a vacuum evaporator (R-20, Buchi, Switzerland) at 45 °C to obtain coffee cherry pulp extract by maceration (CCM), coffee cherry pulp extract by Soxhlet extraction (CCS) and coffee cherry pulp extract by ultrasonic-assisted extraction (CCU). The obtained dry extracts were kept in a tight container and stored at 4–8 °C until processing. The final yields of CCM, CCS and CCU were 19.06 ± 1.24, 18.48 ± 1.87 and 20.84 ± 2.56% *w/w*, respectively.

### 2.3. Chemical Compounds Analysis

#### 2.3.1. Determination of Total Phenolic Content

The total polyphenolic content was determined by employing Folin–Ciocalteu reagent according to a procedure described by Tunit et al. [48]. In total, 50 µL of the coffee cherry pulp extracts in water at a concentration of 25 to 1000 µg/mL were mixed with 100 µL of Folin–Ciocalteu reagent diluted in water at a concentration of 10% *v/v*. After 4 min, 50 µL of 7.5% *w/v* Na_2_CO_3_ in water was added and incubated for 2 h in the dark. Absorbances were measured at 765 nm using a microplate reader (SpectraMax M3, San Jose, CA, USA). Gallic acid at the working concentration of 5 to 120 µg/mL was used to plot the analytical curve, and the results were expressed as mg of gallic acid equivalent per gram of each extract (mg GAE/g).

#### 2.3.2. Determination of Total Flavonoid Content

The flavonoid content of the samples was determined using a modified version of the method described by Tunit et al. [48]. In all, 100 µL of the coffee cherry pulp extract in water at a concentration of 25 to 1000 µg/mL was mixed with 30 µL of 30% *w/v* NaNO_2_ in water and 50 µL of 2% *w/v* AlCl_3_ in water. The solution was left to stand for 6 min, and 1 N NaOH was added to the solution. After incubating for 10 min, the absorbance was measured at 510 nm using a microplate reader (SpectraMax M3, San Jose, CA, USA). The standard curve was constructed with quercetin at the working concentration of 10–400 µg/mL. The flavonoid content was calculated in mg quercetin equivalents per gram of each extract (mg QE/g).

#### 2.3.3. Phytochemical Profile Analysis by HPLC

The quantity of phytochemical profile in the coffee cherry pulp extracts was determined using high-performance liquid chromatography (Shimadzu Prominence, Japan) with modification from Durán et al. [49]. The coffee cherry pulp extracts at a concentration of 1000 µg/mL were dissolved in MeOH and filtered through a 0.45 mm PTFE membrane and finally injected (10 µL) into the HPLC. The C18 column (250 × 4.6 mm i.d.; KNAUER, Vertex III, Berlin, Germany) of the reverse phase HPLC was connected to a UV detector. Acetonitrile, 1% *v/v* acetic acid (15:85) was used to operate the solvent system with a flow rate of 1 mL/min for 20 min, and wavelengths at 280 nm were implemented. The identified compounds were quantified using calibration curves of the standards (chlorogenic acid, caffeine and theophylline). Afterwards, the content of each individual compound was expressed as milligrams per gram dry weight of a coffee cherry pulp (mg/g).

### 2.4. Anti-Oxidant Activity Tests

#### 2.4.1. DPPH Radical Scavenging Assay

The radical scavenging activity of coffee cherry extracts by DPPH assay was determined according to the method of Tunit et al. [48]. First, 100 µL of different concentrations of coffee cherry pulp extracts (25 to 1000 µg/mL) was added to 100 µL of 167 µM DPPH dissolved in ethanol. The mixtures were incubated in the dark at room temperature for 30 min, and the absorbance was measured at 517 nm using a microplate reader (SpectraMax M3, San Jose, CA, USA). Trolox was employed to produce a standard curve. The results were expressed as the Trolox equivalent anti-oxidant capacity (TEAC), which is the concentration of Trolox in micromoles (µM) equivalent to the weight of coffee cherry pulp extracts in grams. 

#### 2.4.2. Lipid Peroxidation Inhibition Using Ferric Thiocyanate Assay

Lipid peroxidation inhibition assay was performed according to the method described by Kiattisin et al. [40], which was modified from Osawa et al. [50]. Different concentrations of coffee cherry pulp extracts (25–1000 µg/mL) in ethanol were mixed with 350 µL of 1.3% *v/v* linoleic acid and 50 µL of AAPH (25.15 mg/mol) in 20 mM PBS (pH 7.0). The final volume was adjusted to 1 mL by adding 20 mM PBS (pH 7.0). After storing at 45 °C for 4 h, 5 µL from each tube was mixed with 5 µL of 10% *w/v* NH_4_SCN and 5 µL of 20 mM FeCl_2_. After incubating for 3 min at room temperature, the absorbance of the mixture was measured at 500 nm using a microplate reader (SpectraMax M3, San Jose, CA, USA). Trolox was employed to produce a standard curve. The results were expressed as the Trolox equivalent anti-oxidant capacity (TEAC), which is the concentration of Trolox in micromoles (µM) equivalent to the weight of coffee cherry pulp extracts in grams. 

### 2.5. Enzymes Inhibitory Tests

#### 2.5.1. Collagenase Inhibitory Assay

Collagenase inhibitory activity was assessed based on the method of Younis et al. [51] with minor modifications. The collagenase enzyme was prepared in 50 mM tricine buffer (pH 7.5) containing 400 mM sodium chloride and 10 mM calcium chloride. The substrate used in this study was a synthetic peptide of FALGPA, which was intended to imitate the structural characteristics of collagen. To prepare the substrate, the synthetic peptide was dissolved in a tricine buffer at a concentration of 2 mM; then, 10 µL of coffee cherry pulp extracts was added to 40 µL of collagenase enzyme solution and incubated at ambient temperature for 15 min. Afterward, 50 µL of the substrate was added to the samples to initiate the reaction. The absorbance was immediately measured at 340 nm in kinetic mode using a microplate reader (SpectraMax M3, San Jose, CA, USA). The absorbance was calculated to express the percentage inhibition of collagenase activity from Equation (1) below:(1)$$\mathrm{Collagenase}\;\mathrm{inhibition}\;(\%)=\frac{{Abs}_{control}-{Abs}_{sample}}{A{bs}_{control}}\times 100$$
where ${Abs}_{control}$ is the absorbance of the reaction with deionized water, collagenase enzyme solution, and the substrate. ${Abs}_{sample}$ is the absorbance of the reaction with extract, collagenase enzyme solution, and the substrate.

#### 2.5.2. Elastase Inhibitory Assay

The elastase inhibition of coffee cherry pulp extracts was conducted using a modified method of Thring et al. [52] to measure the product during the reaction of the elastase enzyme and substrate. Elastase inhibitory assay was performed using a substrate AAAPVN reacted with elastase enzyme. First, 50 µL of coffee cherry pulp extracts was pre-incubated with 25 µL of 2 mg/mL of elastase solution in Tris-HCl buffer (100 mM, pH 8.0) at ambient temperature for 20 min. Then, 25 µL of 4.4 mM of AAAPVN in the buffer was added to initiate the reaction. The absorbance was immediately measured at 410 nm with the kinetic mode using a microplate reader (SpectraMax M3, San Jose, CA, USA). The absorbance was calculated to express the percentage inhibition of elastase activity from Equation (2) below:(2)$$\mathrm{Elastase}\;\mathrm{inhibition}\;(\%)=\frac{{Abs}_{control}-{Abs}_{sample}}{A{bs}_{control}}\times 100$$
where ${Abs}_{control}$ is the absorbance of the reaction with deionized water, elastase enzyme solution, and the substrate. ${Abs}_{sample}$ is the absorbance of the reaction with extract, elastase enzyme solution, and the substrate.

#### 2.5.3. Hyaluronidase Inhibitory Assay

The coffee cherry pulp extracts were evaluated for their ability to inhibit the hyaluronidase enzyme using a modified turbidimetric method from Widowati et al. [53]. The amount of hydrolyzed hyaluronic acid evaluated hyaluronidase enzyme activity. Any residual hyaluronic acid will form turbidity in the final reaction. Altogether, 300 mM phosphate buffer (PBS) at pH 5.35 was prepared for the buffer solution. Firstly, 50 µL of coffee cherry pulp extract was pre-incubated with 100 µL of hyaluronidase enzyme (2 mg/mL) at 37 ± 5 °C for 10 min. Then, 100 µL of 0.03% *w/v* hyaluronic acid in buffer solution was added and re-incubated at 37 ± 5 °C for 45 min. To precipitate the undigested hyaluronic acid, 1 mL of acetic bovine serum albumin, formed from sodium acetate, acetic acid and bovine serum albumin (pH 3.75), was added. The mixture was kept at room temperature for 10 min, and then the absorbance was measured at 600 nm using a microplate reader (SpectraMax M3, San Jose, CA, USA). The percent inhibition of the hyaluronidase activity was calculated from Equation (3) below: (3)$$\mathrm{Hyaluronidase}\;\mathrm{inhibition}\;(\%)=\left(\frac{{Abs}_{sample}}{{Abs}_{control}}\right)\times 100$$
where ${Abs}_{control}$ is the absorbance of the reaction with deionized water, hyaluronic acid solution, and acetic albumin acid solution. ${Abs}_{sample}$ is the absorbance of the reaction with extract, hyaluronidase enzyme solution, hyaluronic acid solution, and acetic albumin acid solution.

#### 2.5.4. Anti-Tyrosinase Activity

The anti-tyrosinase activity was performed according to the method reported by Kiattisin et al. [40]. L-DOPA was used as a substrate. In total, 70 µL of coffee cherry pulp extracts was thoroughly mixed with 70 µL of PBS (pH 6.5) and 70 µL of mushroom tyrosinase enzyme (50 units/mL) in PBS. After incubating at ambient temperature for 10 min, 70 µL of the substrate (2.5 mM L-DOPA) in PBS was added and incubated at ambient temperature for another 20 min. The absorbance was measured at 765 nm using a microplate reader (SpectraMax M3, San Jose, CA, USA). The absorbance was calculated to express the percentage inhibition of tyrosinase activity from Equation (4) below:(4)$$\mathrm{Tyrosinase}\;\mathrm{inhibition}\;(\%)=\frac{{Abs}_{control}-{Abs}_{sample}}{A{bs}_{control}}\times 100$$
where ${Abs}_{control}$ is the absorbance of the reaction with PBS, tyrosinase enzyme solution, and the substrate. ${Abs}_{sample}$ is the absorbance of the reaction with extract, tyrosinase enzyme solution, and the substrate.

### 2.6. Cell Culture 

The human keratinocyte cell line or HaCaT (EP-CL-0090; Elabscience, TX, USA) was grown and maintained in Dulbecco’s Modified Eagle Medium (DMEM) with 10% *v/v* fetal bovine serum supplementation (FBS) and 1% *w/v* penicillin–streptomycin at 37 °C in a humidified atmosphere of 5% *w/v* CO_2_ incubator (Eppendorf, CellXpert C170, USA). In this study, standard reference material of PAH (CRM47930) was used to induce toxicity in human keratinocytes and the formation of ROS. 

### 2.7. Cytotoxicity Test

The MTT assay was employed to evaluate cell viability by measuring the cellular metabolic activity involved in reducing MTT to formazan dye [54]. HaCaT cells were seeded at a density of 1 × 10^4^ cells/well in 96-well plates. After incubating for 24 h, the cells were treated with the extracts and cultured in the absence or presence of PAH for 24 h. The culture media containing 0.5 mg/mL of MTT was added and incubated for 2 h. After discarding the medium, the formazan crystals were dissolved by adding 100 µL of DMSO. The percentage of cell viability was evaluated by measuring the absorbance at 570 nm using a microplate reader (SpectraMax M3, San Jose, CA, USA).

### 2.8. ROS Production Inhibition (H_2_DCFDA Assay)

To investigate the intracellular ROS-scavenging ability of coffee cherry pulp extracts, the intracellular ROS level was quantified using an intracellular ROS probe H_2_DCFDA from the modified method of Zhen et al. [55]. H_2_DCFDA was used to detect the redox status of cells that fluoresced green after PAH-induced oxidative stress in HaCaT cells. The cells were seeded at a density of 5 × 10^5^ cells/mL in 96-well plates and cultured for 24 h. After incubating, the cells were pretreated with the extracts for 2 h before being exposed to PAH (50 µg/mL) in serum-free DMEM for 2 h. The cell cultures were washed with PBS, 2 µM of H_2_DCFDA working solution was added, and incubation continued for another 15 min. The extracted solution was centrifuged at 5000 rpm for 3 min, and the supernatant was used to measure fluorescence intensity using a flow cytometer (BD Accuri™ C6 Plus, Franklin Lake, NJ, USA). The relative intensity was calculated relative to the untreated control.

### 2.9. Statistic Statistical Analysis

The experiments were conducted in triplicate. The data for IC_50_, the phytochemical contents and percentage of enzyme inhibitory activities were analyzed statistically using one-way analysis of variance (ANOVA) followed by Tukey’s HSD test with a *p*-value (*p* < 0.05), which was considered statistically significant. For comparing between treatment groups, Student’s t-test was used for two groups, and one-way ANOVA was used for multiple groups. Data were analyzed using Statistical Package for the Social Sciences (IBM SPSS Statistics 27.0.1 for windows, SPSS Inc., Chicago, IL, USA). Software and data are presented as mean ± S.D. 

## 3. Results and Discussion

### 3.1. Total Phenolic and Total Flavonoid Contents

Phenolic and flavonoid compounds are known for their potent anti-oxidant properties, which are related to the presence of hydroxyl groups and their ability to scavenge radicals. These compounds have the potential to directly enhance anti-oxidant capacity, resulting in protective effects against oxidative damage and promoting health benefits [56,57,58]. To determine the anti-oxidant potential of coffee cherry pulp extracts, the amounts of total phenolics and flavonoids were first quantified. The effect of three different extraction techniques on TPC and TFC in the coffee cherry pulp extracts is shown in Table 1. The total phenol content was determined spectrophotometrically using the Folin–Ciocalteu reagent, and gallic acid was used as a standard. The relationship between gallic acid concentration and absorbance was represented by the equation $y=0.0169x+0.1883,\;\;R^{2}=0.9989\;$(Figure 1a). CCS represented the highest TPC (324.6 ± 1.2 mg GAE/g extract), which was followed by CCU (127.2 ± 0.8 mg GAE/g extract) and CCM (110.0 ± 0.8 mg GAE/g extract), consecutively. The total flavonoid content was determined employing the aluminum chloride method, and quercetin was used as a standard. The relationship between quercetin concentration and absorbance was represented by the equation $y=0.0024x+0.0031,\;\;R^{2}=0.9996\;$(Figure 1b). Similarly, CCS exhibited the greatest TFC (296.8 ± 1.2 mg QE/g extract) compared with the CCU (151.0 ± 1.2 mg QE/g extract) and CCM (136.8 ± 3.7 mg QE/g extract). Among the extracts, CCS, which was obtained by Soxhlet extraction, possessed the highest TPC and TFC. According to Gligor et al. [47], the highest TPC and TFC of ethanolic green coffee bean extract were observed in Soxhlet extraction compared with ultrasound-assisted extraction, turbo extraction and the combination of two techniques. Soxhlet extraction might offer an advantage in terms of exhaustive extraction in that the solvent was repeatedly evaporated, condensed, and re-circulated through the sample until most of the target compounds were extracted. This exhaustive nature of the extraction process contributes to higher TPC and TFC values [59]. These results indicate that Soxhlet extraction was the most efficient method for obtaining the highest total phenolic and flavonoid contents compared with others. These findings suggest that CCS is a rich source of phenolic compounds, and its radical scavenging activity could be attributed to the presence of these constituents.

### 3.2. HPLC Phytochemicals Profiles of Coffee Cherry Pulp Extracts 

To better understand the main individual components in coffee cherry pulp extracts obtained from different extraction methods, the most common components in all extracts were identified as chlorogenic acid, caffeine and theophylline using qualitative and quantitative HPLC. The individual compounds in coffee cherry pulp extracts were identified and quantified by comparing respective retention times and peak areas with pure standard compounds. As shown in Figure 2, chlorogenic acid, caffeine and theophylline were identified as three major individual compounds in coffee cherry pulp extracts. As shown in Table 2, the CCS extract exhibited the significantly highest content of the three main compounds, which were theophylline (45.9 ± 1.0 mg/g), caffeine (45.0 ± 0.8 mg/g), and chlorogenic acid (17.3 ± 0.0 mg/g), respectively. Moreover, CCU and CCM presented a significantly lower amount of theophylline, caffeine and chlorogenic acid than the CCS ranging from 3.6 ± 0.0 to 4.2 ± 0.2, from 26.8 ± 0.0 to 30.0 ± 0.2 and from 6.2 ± 0.1 to 7.3 ± 0.3 mg/g, respectively.

The phytochemicals of coffee cherry pulp extracts vary depending on the extraction method. The HPLC chromatograms revealed that the compound with the highest polyphenol content, specifically chlorogenic acid, was detected in CCS. This result corresponded to the findings of Martínez et al. [60] investigating the phenolic compounds found in fresh coffee pulp. The authors concluded that the highest content of phenolic compounds in the coffee pulp was chlorogenic acid (42.2%). When comparing the alkaloid contents, CCS exhibited the significantly highest levels of theophylline and caffeine. These results suggest that the extraction of chlorogenic acid, caffeine and theophylline with Soxhlet for 60 min is more efficient than extraction at room temperature for 24 h and ultrasonic-assisted extraction for 60 min. Due to the apparatus’s multiple cycles of solvent reflux and percolation along with short extraction time, complete extraction is achievable, resulting in greater quantities of the desired chemicals [47,61]. Therefore, the Soxhlet extraction method exhibited high extraction efficiency, particularly for compounds that are difficult to extract using other methods.

### 3.3. Anti-Oxidant Activities

The anti-oxidant activities of coffee cherry pulp extracts were analyzed using a variety of techniques to determine their mechanisms. To determine whether the extracts have antiradical activity, a DPPH free radical scavenging assay was used to initially screen for antiradical activity with an electron transfer mechanism of the test materials [62]. In these investigations, Trolox was employed as a positive control, whereas chlorogenic acid, caffeine and theophylline were used as a reference standard of coffee cherry pulp extracts. As shown in Table 1, the TEAC values of CCS, CCU and CCM were 98.2 ± 0.8, 52.3 ± 4.3 and 46.2 ± 2.2 µM Trolox/g extract, respectively. Among the extracts, CCS exhibited the highest free radical scavenging activity. Furthermore, there was no significant difference in CCS when compared to the three reference standards of chlorogenic acid, caffeine and theophylline. 

Lipid peroxidation of the skin is a significant phenomenon in the aging process of humans. The ferric thiocyanate assay was used to assess the ability of the extracts to protect lipids from oxidative damage. In the presence of peroxides formed during lipid peroxidation, ferric ions (Fe^3+^) react with thiocyanate ions (SCN^−^) to create a red-colored complex called ferric thiocyanate (FeSCN^2+^) [63]. As shown in Table 1, the obtained results demonstrated that CCS possessed the highest antilipid peroxidation activity with an TEAC value of 136.6 ± 6.2 µM Trolox/g. The following CCU and CCM showed similar TEAC values of 60.3 ± 4.6 and 44.0 ± 3.5 µM Trolox/g, respectively. The potency of all extracts as anti-oxidants against lipid oxidative was lower than the three reference standards of chlorogenic acid, caffeine and theophylline. Chlorogenic acid also showed a significantly higher anti-oxidant effect than caffeine in lipid peroxidation inhibitory activity.

Regarding CCS, it had the greatest anti-oxidant activity as determined by the DPPH and lipid peroxidation inhibition assays, which correlated with the TPC, TFC, and levels of the phenolic compounds and alkaloids. Although high levels of caffeine and theophylline were found in CCS, this class of alkaloids slightly contributes to their anti-oxidant activities. Interestingly, chlorogenic acid was found to contain the highest level in CCS, and it strongly correlated with its anti-oxidant activities. There findings were well supported by Duangjai et al. [38], which revealed that chlorogenic acid and caffeine were the primary constituents of coffee cherry pulp extract. The extract exhibited a significant amount of total phenolic content and showed strong anti-oxidant activity determined by DPPH assay with IC_50_ values of 82 µg/mL. The coffee cherry pulp methanolic extract with a high amount of chlorogenic and caffeine had a strong anti-oxidant capacity with the value of 2.2 mg Trolox equivalents/g dry weight by DPPH assay [39]. Tang et al. [64] also provided strong evidence for the findings of a significant association between the anti-oxidant capacity and TPC in the peel of the Pitahaya fruit. This finding emphasized the individual contribution of several phenolic compounds, including chlorogenic acid, to the overall anti-oxidant capacity of the fruit peel. Moreover, Wu et al. [65] found that the extracts derived from Flos Lonicerae possessed notable anti-oxidant properties with chlorogenic acid identified as a significant contributor to scavenging DPPH radical and reducing Fe^3+^ to Fe^2+^.

Apart from that, the quantity of compounds presented in the extracts was also compared to their standard compounds to assess anti-oxidant activity. The results found that CCS contained 17.3 mg/g of chlorogenic acid and exhibited anti-oxidant activities with a TEAC value of 1.8 µM measured by DPPH. Moreover, CCS contained 45.9 mg/g of theophylline and 45.0 mg/g of caffeine, contributing a TEAC value of 4.5 µM for each. The chlorogenic acid, theophylline and caffeine detected in CCS exhibited lipid peroxidation inhibition with TEAC values of 22.2 µM, 11.0 µM and 25.0 µM, respectively. These results indicate that the whole crude extract exhibits higher anti-oxidant activity than a pure identified compound isolated from it. However, the various unidentified components in the extract may act synergistically. The combined effect is greater than the sum of their individual effects. Different compounds may have complementary anti-oxidant mechanisms, leading to enhanced anti-oxidant capacity.

### 3.4. The Inhibitory Effects on Collagenase, Elastase, Hyaluronidase, and Tyrosinase Enzymes

In terms of the skin aging process, there is an increase in ROS production, which causes damage to the mitochondria in the cell, resulting in an upregulation of matrix metalloproteinases (MMPs) expression in human skin. MMPs, or matrix metalloproteinases, play a crucial role in the buildup of aged cells, decrease in connective tissue, and breakdown of elastic fibers [66]. Therefore, the efficacy of the coffee cherry pulp extracts to inhibit collagenase, elastase, and hyaluronidase enzymes was examined to assess anti-aging properties. All the results were represented as a percentage of collagenase, elastase and hyaluronidase enzymes inhibition at a concentration of 500 µg/mL, as shown in Table 3. 

Collagen serves as the primary component of the skin, imparting structural stability. Collagenases initiate the breakdown of collagen by breaking interstitial collagens. The inhibitory effects of this specific enzyme can help to delay the process of collagen breakdown and the subsequent development of wrinkles [67]. To assess collagenase inhibitory activity, EGCG, a well-known natural collagenase inhibitor [68], was used as a positive control, whereas chlorogenic acid, caffeine and theophylline were used as a reference standard for coffee cherry pulp extracts. The results of collagenase inhibitory activity of coffee cherry pulp extracts indicated that CCS showed the strongest ability to inhibit collagenase enzyme with a value of 89.4 ± 3.8%, which was followed by CCU (75.8 ± 3.7%) and CCM (68.9 ± 3.2%), consecutively. All extracts indicated a higher collagenase inhibitory activity than theophylline and caffeine. Interestingly, CCS did not significantly differ in collagenase inhibitory activity from EGCG and chlorogenic acid. Although the collagenase inhibitory activity of CCM and CCU was not as much as EGCG, CCM and CCU possessed promising anti-collagenase activity. It is worth mentioning that CCS exhibited an inhibitory activity of over 80%, which is equally strong as EGCG. Interestingly, CCS can also inhibit collagenase as effectively as chlorogenic acid. These results are in notable agreement with the study conducted by Mandrone et al. [69], which examined the strong inhibitory capacity of quercetin and chlorogenic acid from *Hypericum hircinum* L. against the collagenase enzyme. The results are well supported by numerous studies reporting that the inhibition of collagenase is mostly attributed to flavonoids and phenolic acids [70,71]. Moreover, Bertolini et al. [72] found that theophylline stimulated an increase in the expression of the anti-oxidant metallothionein-1 protein as well as the messenger RNA (mRNA) levels of collagen I and III. Moreover, theophylline demonstrated extracellular matrix-protective properties through the reduction in mRNA levels of MMP-2 and MMP-9, thereby partially counteracting the effects of menadione, which is a strong and hazardous ROS source.

Elastin is a protein present in connective tissues, serving as the main factor in the regulation of skin elasticity. Elastases, also known as elastase-type proteases, are proteolytic enzymes that can degrade elastic fibers [73]. The inhibition of elastases would delay the degradation of elastin fibers, thereby maintaining the elastic properties of the skin. Therefore, the elastase inhibitory activity was also determined for the coffee cherry pulp extracts. After measuring the activity against the elastase enzyme of the coffee cherry pulp extracts, CCS also possessed the highest ability to inhibit the elastase enzyme with a value of 56.1 ± 1.4%. Additionally, the CCS extract presented significantly greater potential in inhibiting the elastase enzyme compared with caffeine and theophylline. Both CCU and CCM exhibited comparable levels of elastase inhibition with values of 21.3 ± 2.7% and 26.4 ± 2.4%, respectively, which are distinctively lower than those of caffeine and theophylline. However, the inhibitory activity of the coffee cherry pulp extracts on the elastase was not as strong as that observed on collagenase. Polyphenols possess strong anti-oxidant properties that enable them to effectively scavenge ROS according to their chemical composition. Polyphenols can exhibit inhibitory effects on the elastase [71]. Therefore, CCS contained the highest polyphenols and alkaloids contents exhibited the most potent inhibitory effect against elastase. Lee et al. [74] found that caffeine exhibited a significant inhibitory effect on collagenase, followed by elastase, while a less inhibiting effect was shown for tyrosinase in a concentration-dependent manner.

Hyaluronic acid is an essential constituent of the ECM and plays a crucial role in maintaining the optimal moisture levels of human skin. The primary approach by which HA enhancement occurs depends on the suppression of hyaluronidase activity by hyaluronidase inhibitors [75]. As a result of hyaluronidase inhibition, all extracts demonstrated equivalent inhibition of more than 30%. The content of the polyphenols and alkaloids in the coffee cherry pulp extracts may affect the anti-oxidant, anti-collagenase, anti-elastase and anti-hyaluronidase properties of such extracts [76]. However, all test materials were distinctively lower than tannic acid, which was used as a positive control [77].

Melanin is a pigment that causes hyperpigmentation and the production of spots and freckles on the skin. Tyrosinase, an enzyme, is responsible for melanin synthesis in melanocytes [78]. The anti-tyrosinase activity of the coffee cherry pulp extracts is shown in Table 3. The most potent tyrosinase inhibitor was significantly found in CCS compared with other extracts. Kojic acid has been known as a potent anti-tyrosinase agent, which was used as a positive control. Although the tyrosinase inhibitory activity of all extracts was not as much as kojic acid, caffeine and theophylline, CCS possessed promising activity with an IC_50_ value of 523.8 ± 5.1 followed by CCU (890.5 ± 2.4) and CCM (898.0 ± 8.4), consecutively. Considering caffeine and chlorogenic acid, the active compounds presented in the extracts were employed to explain the anti-tyrosinase effect. Similar results were studied by Yang et al. [79] reporting that the caffeine extracted from Camellia pollen had significant inhibitory effects on the intracellular tyrosinase activity and melanin synthesis of B16-F10 cells with the amount of inhibition depending on the dose of caffeine. In this way, it was also indicated that CCS was a potent tyrosinase inhibitor, which is supported by the suggestion of Duangjai et al. [80] that coffee pulp extract containing caffeine and caffeic acid revealed strongly inhibited tyrosinase with 45–47% inhibition at a concentration of 2 mg/mL.

### 3.5. Pearson’s Correlation of Total Phenolic Content, Total Flavonoid Content, and Phytochemicals Content with Anti-Oxidant Activities and Anti-Aging Activities of Coffee Cherry Pulp Extracts 

A correlation analysis was performed among the total phenolic and flavonoid contents and individual phytochemical compound contents measured by HPLC with anti-oxidant capacities measured by DPPH and lipid peroxidation inhibition assays, including the inhibition of skin aging-related enzymes (collagenase, elastase, hyaluronidase and tyrosinase) of coffee cherry pulp extracts (Table 4). The anti-oxidant assays, DPPH and lipid peroxidation inhibition were very strongly correlated (above 0.80) with TPC and TFC. Moreover, high correlation coefficients among TPC, TFC and the inhibition of the collagenase and elastase were observed, suggesting that polyphenols are the main metabolites responsible for contributing to these activities. Moreover, TPC moderately correlated (between 0.40 and 0.59) with hyaluronidase and tyrosinase inhibitory activities, while the strong correlation (0.60 to −0.79) found for TFC suggested that in addition to phenolic compounds, TFC may be contributing to these activities.

Chlorogenic acid, caffeine, and theophylline compounds detected in coffee cherry extracts by HPLC were very strongly correlated with both anti-oxidant activities, suggesting that within the extracts, these compounds significantly contributed to the anti-oxidant activities. Furthermore, the three standard compounds were very strongly correlated to the inhibition of collagenase, elastase and tyrosinase, implying that caffeine and theophylline along with phenolic compounds responded to much of these activities.

### 3.6. Effect of Coffee Cherry Pulp Extracts on PAH-Induced Toxicity in HaCaT Cells

The impact of plant-derived chemicals on cell physiology can be either beneficial or toxic, making the selection of an optimal concentration of the highest priority [81,82]. Considering that coffee cherry pulp extracts contained a mixture of many compounds, the observed results represent the combined effects of the coffee cherry pulp extracts as a whole. The cytotoxicity of coffee cherry pulp extracts on HaCaT cells was assessed by measuring cell viability using the MTT assay. The cells were treated with the extracts at different concentrations (100, 250, 500 and 1000 µg/mL) for 24 h. The results showed that no significant decreases in cell viability were observed at low concentrations of all extracts up to 500 µg/mL but significantly reduced at 1000 µg/mL compared with that observed in the untreated control (Figure 3). Theophylline, a component of coffee cherry pulp extracts, was shown to decrease menadione-stimulated epidermal keratinocyte apoptosis [72]. Various works have reported the reduction in apoptosis and cell viability of chlorogenic acid in various cells, such as osteoblastic cells and human keratinocyte cells [83,84]. Moreover, Gherardini et al. [85] found that the topical application of 0.1% caffeine did not induce skin or hair follicle cell cytotoxicity. Therefore, coffee cherry pulp extracts showed no toxicity to keratinocytes, and chlorogenic acid, caffeine and theophylline were also relatively less toxic. Therefore, 100, 250 and 500 µg/mL of the extracts exhibited no cytotoxicity toward keratinocytes and were chosen for subsequent experiments. 

However, no related study has been conducted concerning the cytoprotective effect of coffee cherry pulp extract against PAH toxicity and the effect of these extracts on oxidative stress induced by PAH. PAH can induce an increase in ROS levels in different cell lines, including human keratinocyte cells [86,87]. To determine whether PAH induced the cytotoxicity of HaCaT cells via MTT assay, the cells were treated with various concentrations of PAH (25, 50, 100 and 200 µg/mL) for 24 h. As shown in Figure 4a, the results showed a gradual increase in PAH in toxicity up to 200 µg/mL. PAH was not toxic for HaCaT cells at concentrations of 25 and 50 µg/mL but significantly decreased cell viability above 100 µg/mL. The concentration of 100 µg/mL of PAH initially had the potential to induce cell toxicity. Comparable findings concerning the harmful effects of PM_2_._5_ by Li et al. [88] showed that the rate of cell damage gradually rose with the rise in PM_2_._5_ concentration. When the concentration of PM_2_._5_ reached 50 µg/mL, the HaCaT cell viability stabilized (around 70%). The majority of the cells died when the concentration of PM_2_._5_ exceeded 100 µg/mL. Therefore, in this study, 100 µg/mL of PAH was chosen as the optimal concentration in this experiment because it generated persistent and significant cytotoxicity. Moreover, PAH at 50 µg/mL was used for subsequent experiments to investigate intracellular ROS generation.

To determine the potential of coffee cherry pulp extracts to protect HaCaT cells against PAH exposure, the cells were treated with CCM, CCS and CCU with various concentrations (100, 250 and 500 µg/mL) and 100 µg/mL of PAH for 24 h. As exposure to the PAH resulted in a 40% decrease in cell viability, CCM treatment was initiated to inhibit the decrease in cell viability at a high concentration range of 250 to 500 µg/mL (Figure 4b). Nevertheless, it was observed that CCS (Figure 4c) and CCU (Figure 4d) exhibited a potent inhibitory effect on the decrease in viability of cells exposed to PAH at lower concentrations, which was notably between the range of 100 and 500 µg/mL. 

### 3.7. Effect of Coffee Cherry Pulp Extracts on Intracellular ROS Level

Flow cytometry was employed to investigate intracellular ROS generation. The oxidant-sensitive probe H_2_DCFDA was used to detect intracellular levels of ROS. According to the data illustrated in Figure 5, the relative fluorescence intensity of the HaCaT cells was compared between cells treated with PAH (50 μg/mL) in the absence and presence of coffee cherry pulp extracts at different concentrations of 100 to 500 μg/mL. The results indicated that the treatment of PAH (50 μg/mL) resulted in a significant increase in intracellular ROS levels as compared with the control group that did not receive any treatment. In comparison, after extracts treatment, only CCS at the lowest concentration from 100 to 500 μg/mL exhibited a considerable ability to mitigate the creation of the ROS generated by PAH without cell disruption from CCS itself (Figure 6). This finding suggests that treatment with CCS has the potential to prevent and suppress the intracellular generation of ROS in HaCaT cells that is caused by exposure to PAH. 

Hyun et al. [89] reported that exposure to UVB radiation and PM_2_._5_ resulted in cellular damage through the mechanism of oxidative stress. The authors observed that the anti-oxidative properties of chlorogenic acid alleviated the detrimental effects of oxidative stress. It has been reported that chlorogenic acid can reduce PM_10_-induced cell death, LDH release and ROS production [84]. The results of both studies agree well with each other. Although chlorogenic acid exhibits strong intracellular ROS scavenging activity exposed by PMs, theophylline exhibited a reduced apoptosis of epidermal keratinocytes induced by menadione. Notably, theophylline exhibited the ability to protect skin cells against ROS and repair DNA damage [72]. Furthermore, caffeine can protect human skin from fibroblast cell damage from an intermediate dose of H_2_O_2_ (0.1 mM) [90]. It has been presumed that chlorogenic acid possesses a higher probability of acting as an efficient anti-oxidant and protective effect from PAH-induced oxidative compared with caffeine and theophylline. Overall, our findings confirmed that the Soxhlet extraction method demonstrated a high potential for obtaining bio-active compounds with high anti-oxidant, anti-aging activities and was more effective for protecting the skin from air pollutants. 

## 4. Conclusions

The impact of extraction techniques on the phytochemistry and biological activities of coffee cherry pulp extracts was studied. Among the obtained extracts, CCS extract possessed the most desirable biological activities. It exhibited the greatest anti-radical activity, as proven by the DPPH assay, anti-oxidant activity as proven by lipid peroxidation inhibition assay and anti-aging activities as proven by collagenase, elastase, hyaluronidase and tyrosinase inhibition compared with the other investigated extracts. The cell culture experiments indicated that the CCS extract was non-cytotoxic toward human keratinocyte cells. The measured ROS activity showed that CCS can reduce oxidative stress and increases cell viability in PAH-exposed conditions. Chlorogenic acid, caffeine and theophylline enriched in CCS may constitute active phytochemicals providing anti-oxidant, anti-aging and cytoprotective effects. The properties above revealed in this study allowed us to suppose that CCS has multifunctional activities. For this reason, CCS will be subjected to further in vivo study to confirm its benefits for cosmeceutical application. This approach could contribute to developing natural ingredients with anti-aging and protective effects from air pollution.

## Figures and Tables

**Figure 1 foods-12-04292-f001:**
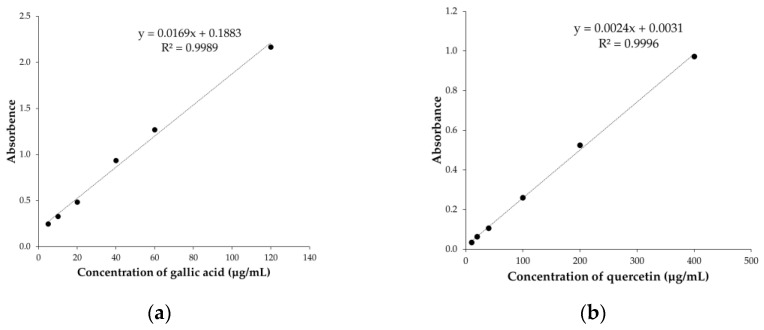
(**a**) Calibration curve of gallic acid; (**b**) calibration curve of quercetin.

**Figure 2 foods-12-04292-f002:**
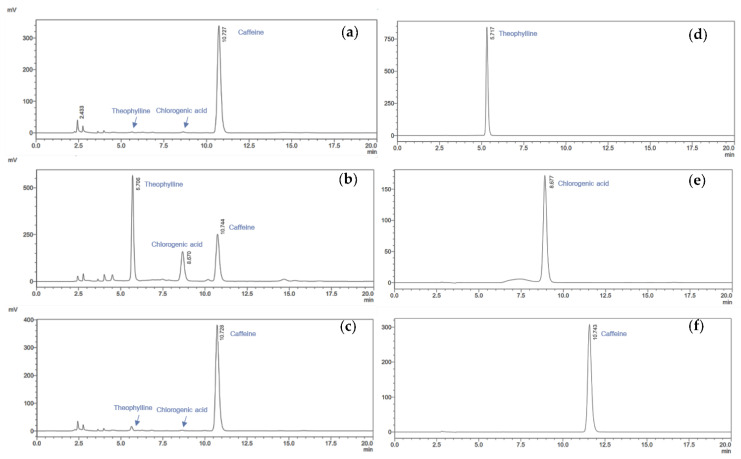
HPLC chromatograms of the coffee cherry pulp extracts: (**a**) CCM, (**b**) CCS, and (**c**) CCU at a concentration of 1000 µg/mL and standard substances of (**d**) theophylline, (**e**) chlorogenic acid, and (**f**) caffeine at a concentration of 120 µg/mL were detected at 280 nm using C18 reversed phase column. The analysis was performed using the isocratic mixture of acetonitrile and 1% *v/v* acetic acid (15:85) with a flow rate of 1 mL/min for 20 min.

**Figure 3 foods-12-04292-f003:**
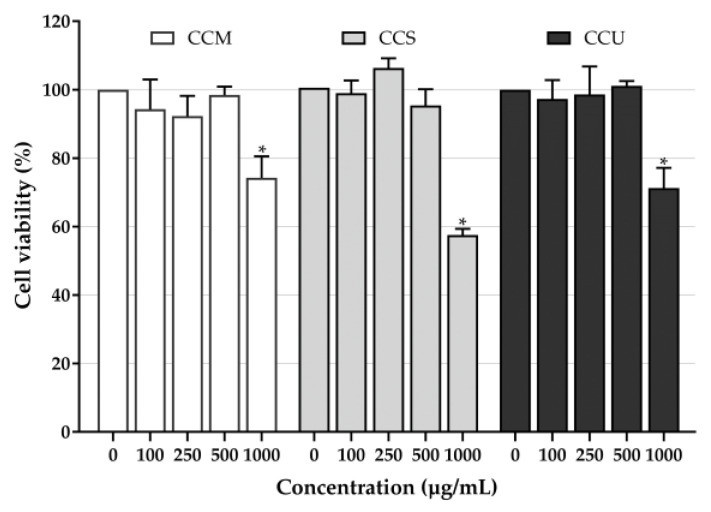
Effects of coffee cherry pulp extracts on the viability of HaCaT cells using the MTT assay; the bars represent the percent cell viability of HaCaT cells treated with different concentrations of the coffee cherry pulp extracts (100–1000 µg/mL) for 24 h. Data are shown as mean ± S.D. (*n* = 3). Asterisk (*) presents significant differences compared with the untreated cells (0 µg/mL) at *p <* 0.05.

**Figure 4 foods-12-04292-f004:**
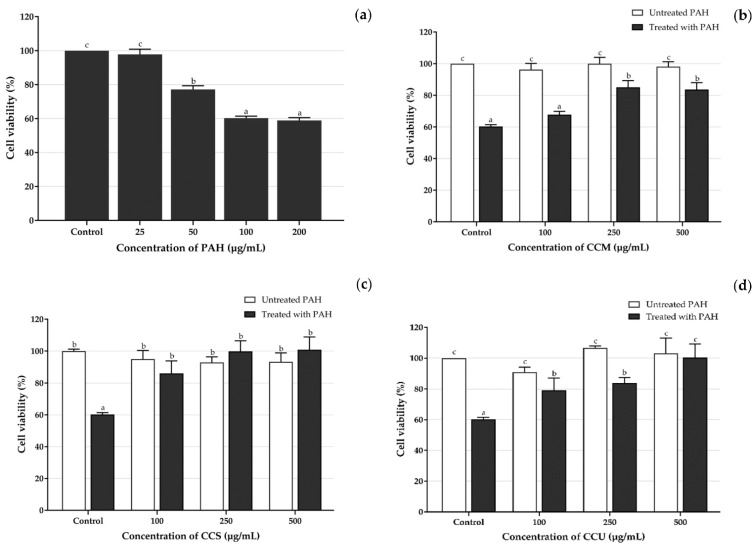
Effects of PAH on the viability of HaCaT cells using the MTT assay; the bars represent the percent cell viability of HaCaT cells (**a**) treated with different concentrations of the PAH (25–200 µg/mL) and the extracts; (**b**) CCM, (**c**) CCS and (**d**) CCU in different concentrations (100–500 µg/mL) were exposed to 100 µg/mL of PAH for 24 h. Data are shown as mean ± S.D. (*n* = 3). Tukey’s HSD test was performed to compare all group means to each other. Groups that share the same letters (**a**–**c**) do not exhibit significant differences at *p <* 0.05.

**Figure 5 foods-12-04292-f005:**
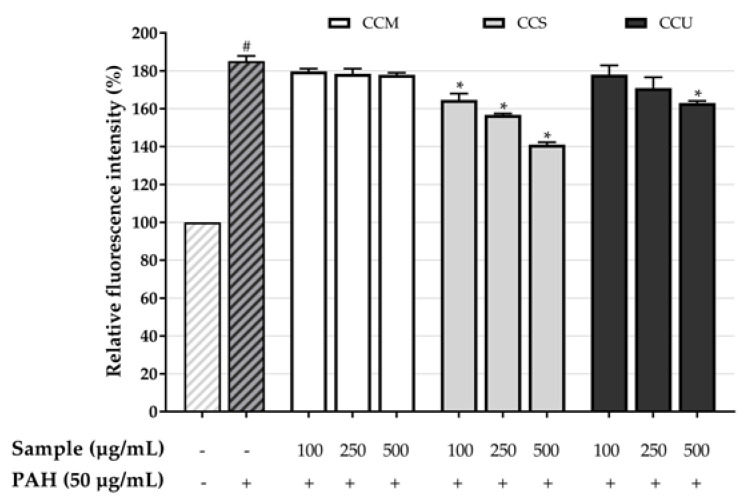
Percentage of relative intracellular ROS fluorescence intensity of coffee cherry pulp extracts at different concentrations of 100–500 μg/mL. The # *p* < 0.05 was significant compared with untreated cells (0 µg/mL). The * *p* < 0.05 was significant compared with the PAH-only control.

**Figure 6 foods-12-04292-f006:**
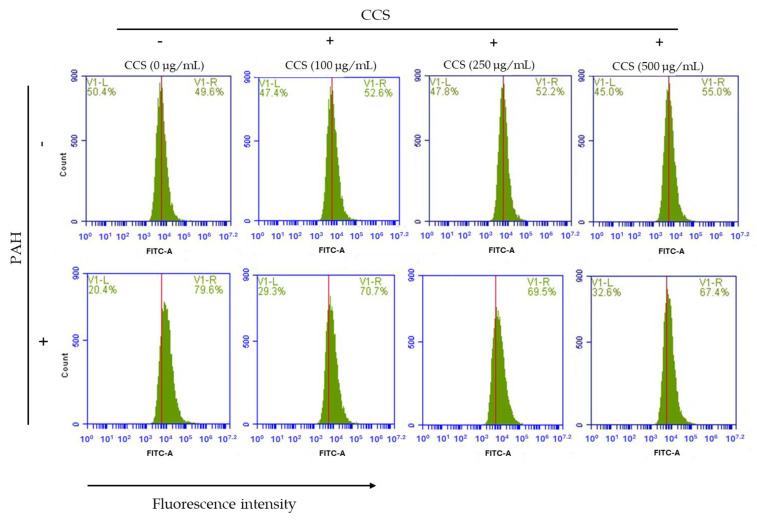
Histograms of the HaCaT cell counts versus fluorescence intensity are shown with a mark to define fluorescing cells of CCS at different concentrations of 100–500 μg/mL.

**Table 1 foods-12-04292-t001:** Total phenolic content, total flavonoid content, and anti-oxidant activities (DPPH and lipid peroxidation assays) of the coffee cherry pulp extracts compared to chlorogenic acid, caffeine and theophylline standards.

Test Material		TPC (mg GAE/g Extract)	TFC (mg QE/g Extract)	Anti-Oxidant Activity TEAC (µM Trolox/g)	
				DPPH	Lipid Peroxidation Inhibition
Coffee cherry pulp extract	CCM	110.0 ± 0.8 ^a^	136.8 ± 3.7 ^a^	46.2 ± 2.2 ^a^	44.0 ± 3.5 ^a^
	CCS	324.6 ± 1.2 ^c^	296.8 ± 1.2 ^c^	98.2 ± 0.8 ^b^	136.6 ± 6.2 ^b^
	CCU	127.2 ± 0.8 ^b^	151.0 ± 1.2 ^b^	52.3 ± 4.3 ^a^	60.3 ± 4.6 ^a^
Standard	Chlorogenic acid	NT	NT	104.2 ± 0.6 ^b^	1283.8 ± 37.5 ^e^
	Caffeine	NT	NT	100.6 ± 0.9 ^b^	556.5 ± 25.8 ^d^
	Theophylline	NT	NT	97.5 ± 3.1 ^b^	240.3 ± 8.6 ^c^

Note: CCM = coffee cherry pulp extract by maceration, CCS = coffee cherry pulp extract by Soxhlet extraction, and CCU = coffee cherry pulp extract by ultrasonic-assisted extraction. NT = not tested. Each value represents the mean ± S.D. (*n* = 3), and different letters within the same column indicate significant differences (*p* < 0.05) between groups according to Tukey’s HSD one-way ANOVA.

**Table 2 foods-12-04292-t002:** Quantification of compounds in the coffee cherry pulp extracts by HPLC.

Test Material	Amount (mg/g Extract)		
	Theophylline	Chlorogenic Acid	Caffeine
CCM	4.2 ± 0.2 ^a^	6.2 ± 0.1 ^a^	26.8 ± 0.0 ^a^
CCS	45.9 ± 1.0 ^b^	17.3 ± 0.0 ^c^	45.0 ± 0.8 ^c^
CCU	3.6 ± 0.0 ^a^	7.3 ± 0.3 ^b^	30.1 ± 0.2 ^b^

Note: CCM = coffee cherry pulp extract by maceration, CCS = coffee cherry pulp extract by Soxhlet extraction, and CCU = coffee cherry pulp extract by ultrasonic-assisted extraction. Each value represents the mean ± S.D. (*n* = 3), and the different letters within the same column indicate significant differences (*p* < 0.05) between groups according to Tukey’s HSD one-way ANOVA.

**Table 3 foods-12-04292-t003:** Ability of the coffee cherry pulp extracts to inhibit collagenase, elastase, hyaluronidase and tyrosinase enzymes compared with positive controls and reference standards at a concentration of 500 µg/mL.

Test Material		Inhibition (%)			Tyrosinase (IC_50_; µg/mL)
		Collagenase	Elastase	Hyaluronidase	
Coffee cherry pulp extract	CCM	68.9 ± 3.2 ^b^	26.4 ± 2.4 ^a^	31.1 ± 3.3 ^b^	898.0 ± 8.4 ^e^
	CCS	89.4 ± 3.8 ^c^	56.1 ± 1.4 ^c^	32.5 ± 1.7 ^b^	523.8 ± 5.1 ^d^
	CCU	75.8 ± 3.7 ^b^	21.3 ± 2.7 ^a^	35.9 ± 0.9 ^b^	890.5 ± 2.4 ^e^
Positive control	EGCG	95.9 ± 3.8 ^c^	71.1 ± 1.7 ^d^	NT	NT
	Tannic acid	NT	NT	98.8 ± 1.8 ^d^	NT
	Kojic acid	NT	NT	NT	25.5 ± 0.5 ^a^
Standard	Chlorogenic acid	85.4 ± 5.2 ^c^	69.7 ± 6.6 ^d^	50.1 ± 1.0 ^c^	121.0 ± 1.4 ^b^
	Caffeine	39.0 ± 1.3 ^a^	34.0 ± 3.2 ^b^	19.1 ± 1.4 ^a^	130.0 ± 3.6 ^b^
	Theophylline	37.6 ± 1.0 ^a^	35.4 ± 3.6 ^b^	19.7 ± 2.1 ^a^	165.9 ± 5.5 ^c^

Note: CCM = coffee cherry pulp extract by maceration, CCS = coffee cherry pulp extract by Soxhlet extraction, and CCU = coffee cherry pulp extract by ultrasonic-assisted extraction. NT = not tested. Each value represents the mean ± S.D. (*n* = 3), and the different letters within the same column indicate significant differences (*p* < 0.05) between groups according to Tukey’s HSD one-way ANOVA.

**Table 4 foods-12-04292-t004:** Pearson’s correlation coefficients (r) among the TPC, TFC, and phytochemicals contents with anti-oxidant and enzyme inhibitory activities in coffee cherry pulp extracts.

	Variable	Anti-Oxidant Activity		Enzyme Inhibitory Activity			
		DPPH	Lipid	Collagenase	Elastase	Hyaluronidase	Tyrosinase
Coffee cherry pulp extracts	TPC	0.900 **	0.915 *	0.790 *	0.906 *	0.552	0.565
	TFC	0.930 *	0.881 *	0.898 *	0.977 **	0.746	0.723
The detected amount of compound in the extracts							
Chlorogenic acid	HPLC	0.958 **	0.891 *	0.969 **	0.975 **	0.309	1.000 **
Caffeine	HPLC	0.934 *	0.922 *	0.985 **	0.956 **	0.379	0.998 **
Theophylline	HPLC	0.982 **	0.842*	0.940 *	0.992 **	0.214	0.994 **

** Significant correlation with *p <* 0.01, * Significant correlation with *p <* 0.05.

## Data Availability

Data is contained within the article.

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
