# Peer review of "Comparison of Biological Activities and Protective Effects on PAH-Induced Oxidative Damage of Different Coffee Cherry Pulp Extracts"

_foods, 2023, doi:10.3390/foods12234292_

Round 1

Reviewer 1 Report

Comments and Suggestions for Authors

The manuscript compared the total phenolic content, total flavonoid content, antioxidant activities, anti-tyrosinase activity of coffee cherry pulp extracts from different extraction methods. Generally, it is an interesting work, however, there are several points that need to be addressed:

Title: the title does not appropriately describe the whole manuscript. Actually, the extracts have not been added to cosmetic material and they did not really detect its anti-aging effects on humans' skin. I suggest the authors change the title to be more specific. 

Introduction: in Lines 121-122, the authors claimed that the Soxhlet as the green extraction technique, but I don't agree with it. In several kinds of literature and it is very common to recognize that Soxhlet is very time-consuming and not environmentally friendly, as it is always accompanied by a long heating time (energy consumption) and post-organic solvent treatment. Please consider revising it. 

The novelty of the current manuscript should be added to the manuscript at the end of the introduction.

Material and methods: Equations 1-6: Please change the symbol in the equations. All are described as "A" and "B" in different equations, which makes the reader very easily confused. 

The version, company, and country of the software should be provided.

Please combine the results and discussion together to avoid repetitive descriptions and make it easier to follow.

Eq. 5, Lines 334 and 339: please use the equation form. 

Tables 1 and 3 are quite confusing. I am not sure what the points of measuring "Trolox,  chlorogenic acid, caffeine, theophylline", are they the standard? Abbreviations in the tables should be explained as the note under the table. Please consider to re-organize the table. 

Comments on the Quality of English Language

English is fine but tables 1 and 3 are quite hard to understand

Author Response

Dear Editor of Foods,

Dear Reviewer,

We greatly appreciate the valuable comments and suggestions from all reviewers. We have carefully read and responded to all comments, point by point. The specific alterations in the manuscript in response to the reviewer's comments are shown in yellow highlights.

We hope all of the changes have addressed the reviewers’ concerns, so with these additions, we hope our work will be accepted for publication in Foods.

Best regards,

Asst. Prof. Dr. Kanokwan Kiattisin

Reviewer 2 Report

Comments and Suggestions for Authors

Abstract:

L.20: delete or use commas: "..., which is by-products in the coffee industry,..."

L.22: should be "the extract of coffee cherry pulp"

L.30: "The results demonstrated that CCS could extract..." Instead of CCS, in my opinion, it should be "Soxhlet", in other cases it doesn't make sense.

Keywords are not well-defined, they cannot be the same phrase as the title e.g. "coffee cherry pulp extract" (this is even too long; or "extraction" (maybe "types of extraction" or more directly "Soxhlet extraction"); not "antioxidant" but "antioxidative compounds".

Introduction

L.100 and L.104 I suggest deleting the values and using them in the discussion part.

I advise to pool 2.2.1, 2.2.2 and 2.2.3 into one paragraph. And try to work on a better phrasing of "coffee cherry pulp extract solution". Any abbreviation for this? and possibly delete the extract or solution?

Line 198: should be "was" instead of "is". Extensive English editing is required. Moreover, The description of methods should be shortened and only the most important information should be shown. For example, the equation for the DPPH method should be omitted, it is a well-known method so the Authors don't show any novelty here. Also, if there were no modifications to the method, there is no necessity to present the whole methodology. 

L.115 I don't see a novelty of this research if maceration, Soxhlet and ultrasonic-assisted extraction have been already used. Any modifications in this manuscript?

I advise adding the whole profile of phytochemicals in the Supplementary materials. Moreover, the discussion part in half sounds like an introduction, only with a very brief comparison to the literature. Also, below each table, it could be good to add the meanings of abbreviations: CCM, CCS, and CCU.

It could be a good idea to have a look at the investigations about coffee silver skin, as it is also a waste.

Comments on the Quality of English Language

English corrections are required.

Author Response

Dear Reviewer,

We greatly appreciate the valuable comments and suggestions from all reviewers. We have carefully read and responded to all comments, point by point. The specific alterations in the manuscript in response to the reviewer's comments are shown in yellow highlights.

We hope all of the changes have addressed the reviewers’ concerns, so with these additions, we hope our work will be accepted for publication in Foods.

Best regards,

Asst. Prof. Dr. Kanokwan Kiattisin

Reviewer 3 Report

Comments and Suggestions for Authors

The provided manuscript is devoted to studies of coffee pulp extracts with a plausible application for cosmetic purposes. The manuscript is well structured, however the reviewer has noticed some places which could be improved.

1.       The title is really long (more than 3 lines). It gives full information about    the study, however it could no be considered as eye-catching. Maybe authors could provide some shorter version.

2.       The abstract. It would be excellent to add the Latin name for coffee pulps. The abstracts is rather long: I would suggest to eliminate the sentences e.g., regarding the used test methods and to leave only the results. E.g. eliminate the text that antioxidant activity was measured by DPPH, just leave the info about the obtained results.

3.       Page 2, line 55: polycyclic… should not be in capital

4.       Page 2, line 84-85: sentence ‘’caffeine….’’ Is not clear and should be changed.

5.       Page 3, the last paragraph in the introduction: ‘’ in this study…’’ partially shows the results obtained in the paper. Here should be written your goal, not results / conclusions

6.       Regents: 6-hydroxy-2,5,7,8… should be written with capital H

7.       Reagents, page 3, the last line- all other chemicals… what are these other chemicals?

8.       Page 4: Plant preparation: the method about obtaining pulp is not clear. It should be added.

9.       Page 4: Soxlet extraction: for this method gentle boiling is required. Thus the temperature seems a little confusing.

10.   For all analytical methods the solvent for each solution should be specified.

11.   Page 5. DPPH test. What was the solvent for the test? It is a crucial information in context with the plausible mechanism (it is mentioned some pages later)  

12.   Enzyme inhibitory tests: I would suggest to add the information about the isolation of the enzyme in the ‘’reagents’’ part. It is disturbing at the enzyme tests

13.   Carefully should be checked if information about the number of analyzed samples is added.

14.   Page 8 and next pages: the numbers after . in decimal values: is it really useful to have two digits? Maybe one is enough or even none?

15.   About DPPH and lipid peroxidation tests: I consider that the results should be analyzed also with the respect of concentration of e.g. chlorogenic acid. For the moment the results of extracts seems remarkably worth in comparison with pure compounds. But if the amount of the compounds would be taken into consideration they could appear more similar. (e.g. CCS contains only 17 mg/g chlorogenic acid (less than 2%). IC50 for CCS is 182 mkg/mL and only 2% from this weight is from chlorogenic acid). I suggest the authors to check the data also from the viewpoint of the concentration of active ingredient. Another alternative could be to express all data as equivalents of some reference antioxidant. Such recheck of the data could be valuable for the discussion of all data.

16.   Page 8. Antioxidant activity: to determine whether the extracts have… I cannot agree with this statements. Already in early 2000-thies hard discussions regarding the mechanisms and reaction media  have risen. Please check the work of Litwinienko and Ingold. An information on the reactions media for the DPPH test is missing, but I believe that the reaction was run in ethanol. If it is true than the SPLET mechanism could be more reasonable.

17.   Page 9: the quality of chromatograms should be improved. The presentation of the chromatograms for the extracts is logical, but I do not consider that chromatograms of the standard compounds should be incorporated. The legend of Fig. 1. Should be accompanied by the chromatographic conditions.

18.   Page 9: lipid peroxidation results: is the amount of the chlorogenic extract standardized in the extract? The same discussion here above at point 15.

19.   Page 10: 2nd paragraph somewhat strange spaces are used.

20.   Table 4: chlorogenic acid HPLC; caffeine HPLC… the presentation of the information should be reconsidered. After several readings I got the idea, but the first thought was that the data correspond to pure compounds not the amount of the compound in the extract.

21.   Fig 2. The colours of the bars are really similar. Some differences would be helpful.

22.   Fig 3. The meaning of the colour for the bars is not clear. No explanation in some legend.

23.   Fig 4. Too similar colours for the bars.

24.   Fig 5. The quality of the figure should be improved.

25.   Page 14. Do you have any explanation why TPC and TFC was the highest for the Soxlet extract? Extraction in US or MW is considered as a fast and effective alternative

26.   Terms ‘’antioxidant activity’’  and ‘’antiradical activity’’ are used for different results. Here it should be checked and the correct terms should be used more careful.

27.   What is the yield of various extracts? How the extracts were stored prior the analysis?

Author Response

(The authors gave the same response as above.)

Reviewer 4 Report

Comments and Suggestions for Authors

Dear Authors and Editors,

the article deals with an important and practical aspect of side stream valorization, namely the re-use of cherry pulp extracts. The research uses multiple assays for the analysis of compounds (HPLC-PDA-MS/MS) and bioassays, and antioxidant assays.

Basically the results are well presented and documented, however there are also some things to correct:

After reading it several times the title is still confusing and too condensed to be understood. After reading the work it is OK but authors should somehow fimplify the title to be more understandable.

Section 2.2.2. For Soxhlet extraction how much liquid was used for extraction? PLease clarify.

Please explain why you used 60 min of extraction for Soxhlet! Usually for Soxhlet we take 20-30 cycles for complete extraction, which means about 10-24 hours of extraction. By extracting for 60 min it means that you had cycle times of 2-3 minutes. Is this OK? IF you really extracted for 60 min then the extraction does not seem to be completed.

Section 2.2.3: How did you assure that you kept between 50 +/- 5 C for ultrasonication? Did you use some temperation? Please clarify.

Section3:

L333-339 on calibration seem to belong to materials and methods section as these are no results but methodological details. 

Comments on the Quality of English Language

English is OK, but check with a native speaker would be good.

Author Response

(The authors gave the same response as above.)

Round 2

Reviewer 1 Report

Comments and Suggestions for Authors

Although parts of my comments are addressed, there are still some not been modified. 

For Eqs. 1-4, it is not a big deal to just change the characters. As it may make the readers confused about which equations are.

For the combination of results and discussion, it indeed makes the reader easy to follow, although the template is separated, there are several papers are combined them together: please refer to: "Effect of Extraction Methods on Aroma Profile, Antioxidant Activity and Sensory Acceptability of Specialty Coffee Brews", "Hair Growth Promotion and Anti-Hair Loss Effects of By-Products Arabica Coffee Pulp Extracts Using Supercritical Fluid Extraction" "Evaluation of Physicochemical Characteristics and Sensory Properties of Cold Brew Coffees Prepared Using Ultrahigh Pressure under Different Extraction Conditions".

The line mentioned in the reply to the referee is inconsistent with the modified parts, please make it consistent when you reply. It wastes the reviewers' time to find where the answers are in the revised version. 

Author Response

Dear reviewer,

We apologize for the incomplete previous responses. Thank you,  reviewer, for giving us the opportunity to improve the quality of our manuscript. We greatly appreciate the valuable comments and suggestions from all reviewers. We have carefully read and revised the manuscript to verify our responses to all comments, point by point. The specific alterations in the manuscript in response to the reviewer's comments have been tracked and shown in different highlights.

We hope all of the changes have addressed the reviewers’ concerns, so with these additions, we hope our work will be accepted for publication in Foods.

Best regards,

 Asst. Prof. Dr. Kanokwan Kiattisin
